



# The composition of endolithic communities in gypcrete is determined by the specific microhabitat architecture

María Cristina Casero[1*], Victoria Meslier[2$], Jocelyne DiRuggiero[2], Antonio Quesada[3], Carmen Ascaso[1], Tomasz Kowaluk[4], Jacek Wierzchos[1**]

[1]Departamento Biogeoquímica y Ecología Microbiana, Museo Nacional de Ciencias Naturales, CSIC, Madrid, 28006, Spain
[2]Department of Biology, and Department of Earth and Planetary Sciences, Johns Hopkins University, Baltimore, MD, 21218. USA
[3]Departamento de Biología, Universidad Autónoma de Madrid, Madrid, 28014, Spain
[4]Institute of Metrology and Biomedical Engineering, Faculty of Mechatronics, Warsaw University of Technology, 02-525 Warsaw, Poland
$ now at: MetaGenoPolis, Jouy-en-Josas, France

*Correspondence to*: María Cristina Casero* (mcristina.casero@mncn.csic.es) and Jacek Wierzchos** (j.wierzchos@mncn.csic.es)

**Abstract.** Endolithic microhabitats have been described as the last refuge for life in arid and hyper-arid deserts where life has to deal with harsh environmental conditions. A number of rock substrates from the hyper-arid Atacama Desert, colonized by endolithic microbial communities, such as halite, gypsum crusts, gypcrete, calcite, granite and ignimbrite, have been characterized and compared using different approaches. In this work, three different endolithic microhabitats are described, each one with a particular origin and architecture, found within a lithic substrate known as gypcrete. Gypcrete, an evaporitic rock mainly composed of gypsum ($CaSO_4 \cdot 2H_2O$) and collected in the Cordón de Lila area of the desert (Preandean Atacama Desert), was found to harbour cryptoendolithic (within pore spaces in the rock), chasmoendolithic (within cracks and fissures) and hypoendolithic (within microcave-like pores in rock-bottom layer) microhabitats. A combination of microscopy investigations strategies and high-throughput sequencing approaches were used to characterize the endolithic communities at the microscale in these microhabitats within the same piece of lithic substrate. Microscopy techniques revealed differences in the architecture of the endolithic microhabitats and in the distribution of the microorganisms within those microhabitats. Cyanobacteria and Proteobacteria were dominant in the endolithic communities, of which the hypoendolithic community was the least diverse and hosted unique taxa. These results show, for the first time, that the differences in the architecture of a microhabitat, even within the same piece of lithic substrate, might be an essential factor in shaping the diversity and composition of endolithic microbial communities.



## 1. Introduction

The statement developed by Professor Lourens Gerhard Marinus Baas-Becking (1934) "everything is everywhere but the environment selects" which established the most referred principle for microbial biogeography remains in discussion regarding the first half of the statement (everything is everywhere) (de Wit and Bouvier 2006, O'Malley 2008, Bass and Boenigk 2011, Fontaneto and Hortal 2012, van der Gast 2015). Regarding the second half of the statement (but, the environment selects) extreme environments present some of the most noticeable scenarios since they are inhabited only by microorganisms, which

are able to survive and/or thrive in their respective physical or geochemical extreme conditions, such as: temperature, solar radiation, pressure, desiccation, pH (Rothschild and Mancinelli 2001).

Hyper-arid deserts, where aridity index is lower than 0.05 (Nienow, 2009) constitute the most extreme deserts on Earth, and usually combine a series of simultaneous stress conditions such as water stress, extreme high and low temperatures, scarcity of organic carbon, high solar radiation and osmotic stress (Pointing and Belnap 2012). While these environments are considered

polyextreme, they are inhabited by microbiota able to survive in such extreme conditions. Hence, polyextreme environments are excellent microbial ecosystem models to study adaptive mechanisms to environmental stress. Among others deserts, the Atacama Desert (North Chile) is perhaps the most challenging polyextreme environment on Earth and the most barren region imaginable, with scarce precipitations (McKay et al. 2003; Wierzchos et al. 2012a) and an extremely low mean annual relative humidity (RH) (Azúa-Bustos et al. 2015). Further, this desert holds another world record: the highest surface ultraviolet

radiation (UV), photosynthetic active radiation (PAR) and annual mean surface solar radiation (Cordero et al. 2018).

In this inhospitable polyextreme desert, microbial life has found refuge in very specific endolithic (inside rocks) microhabitats (rev. by Wierzchos et al. 2018; Wierzchos et al. 2012b). Three different locations of these endolithic habitats have been described within rocks of the Atacama Desert: cryptoendolithic (occupying pore spaces in the rock), chasmoendolithic (living within cracks and fissures in the rock), and hypoendolithic (living inside the rock but close to the bottom). Endolithic

colonization can be viewed as a stress avoidance strategy whereby the overlying mineral substrate provides protection from incident lethal UV and PAR radiation, and also offers enhanced moisture availability (Walker and Pace 2007; Wierzchos et al. 2012b). These microbial communities, regardless of the position they occupy in the rock, or the type of rock, are supported by oxygenic phototrophic primary producers supporting a diversity of heterotrophic microorganisms (rev. in Wierzchos et al. 2018). Molecular and microscopy characterization of these endolithic microbial communities shows that, overall, these

communities are dominated by Cyanobacteria, mostly from the extremely resistant to ionizing radiation and desiccation *Chroococcidiopsis* genus (Meslier et al. 2018, Crits-Christoph et al. 2016, Billi et al. 2000, Cockel et al. 2005), and Actinobacteria, Proteobacteria, Chloroflexi, Bacterioidetes and Euryarchaeota phyla (Meslier et al. 2018). In gypcrete and gypsum crust from preandean Atacama Desert, previous studies reported endolithic communities dominated by Cyanobacteria (36-83%), Actinobacteria (10-25%) and Proteobacteria (13-30%) phyla (Wierzchos et al. 2015; Dong et al. 2007; and Meslier

et al. 2018; rev. in Casero et al. 2020), however, these studies did not differentiate between microhabitats, even though the occurrence of different endolithic microhabitats in gypcrete had already been described (Wierzchos et al. 2015).



This work addresses the impact of microhabitat architecture in the diversity and composition of gypcrete endolithic microbial communities (EMCs). The study is based on the hypothesis that the differential architecture of endolithic microhabitats involves small-scale differences in the micro-environmental conditions, which in turn determine the distribution of organisms

in each community. The hypothesis is tested here by using a multidisciplinary approach combining microscopy and molecular tools for their characterization. The microscale dimension and peculiar diversity distribution in this unique environment has led us to coin the new term "microbiogeography".

## 2. Experimental procedures

### 2.1 Site description and sampling

Colonized rocks were collected in the Atacama Desert in December 2015 from the Monturaqui area (MTQ) (GPS coordinates 23°57'S; 068°10'W; 2868 m.a.s.l.) located in a N-S trending depression of the Cordón de Lila Range. This area exhibits a pronounced rain shadow effect by the western slope of the central Andes from 15° to 23°S (DiRuggiero et al. 2013; Wierzchos et al. 2015). In order to study endolithic communities that inhabit the same piece of lithic substrate, we looked for gypcrete pieces that harboured at least two of the three endolithic microhabitats of interest, that were collected within a 50 m$^2$ area. All

samples were packed in sterile bags and stored at room temperature, dry and dark environment before further processing.

### 2.2 Microclimate data

Microclimate data (Meslier et al. 2018) were recorded using an Onset HOBO® Microweather Station Data Logger (H21-USB), as previously described by Wierzchos et al. (2015). Air temperature (T), air relative humidity (RH in %) and Photosynthetic Active Radiation (PAR in μmol photons m$^{-2}$ s$^{-1}$) were recorded from January 2011 to February 2013 (22

months) (Wierzchos et al. 2015). Rainfall data were obtained from DiRuggiero et al. (2013). Thermal measurements of the gypcrete surface were acquired at zenith time at 20 cm distance from the substrate. Thermal images were taken using a thermal infrared camera (FLIR® E6, FLIR Systems, Oregon, USA) whose technical specifications are: ±2°C or ±2% of reading; < 0.06°C pixel sensitivity with resolution of 160 × 120 pixels.

### 2.3 Microscopy analyses

Colonized gypcrete samples were processed for scanning electron microscopy in backscattered detection mode (SEM-BSE) according to methods described by Wierzchos and Ascaso (1994) and Wierzchos et al. (2011). Light microscopy (LM) in differential interference contrast mode (DIC) was performed on cell aggregates gently isolated from the cryptoendolithic, chasmoendolithic and hypoendolithic microhabitats and on cyanobacterial isolated cultures from those microhabitats. The samples were examined using a microscope AxioImager M2, Carl Zeiss, Germany in DIC mode equipped with Apochrome

63x n=1.4 oil immersion objective.



### 2.4 CT-Scan analysis

Micro-CT scans were run on a pieces of gypcrete with an X-Ray Computed Tomography system (CT-scan) — HMXST 225 micro-CT system (Nikon Metrology, Tring, UK) to observe volume, bulk density, and variations in internal density. For volume and bulk density measurements a Nikon X-Tek CT-Scan device was used, with an X-ray peak voltage of 146 kV and

current of 65 mA, collecting 1583 sections at 1000 micro-seconds on average from four frames. The system operates with an X-ray tube and added filtration (0.875 mm Cu) to reduce the beam hardening. Three dimensional viewing and analyses of the obtained X-ray sections were performed by software VG Studio Max Version 2.2. The auto-threshold feature determined the grey-scale intensity for 3-D surface segmentation and subsequent analysis.

### 2.5 Cyanobacteria isolation and characterization

Scrapped material from endolithic colonization zones of gypcrete was transferred to BG11 1.5%-agar plates (Purified agar, Condalab, Spain). All samples were incubated in growth chamber at 28±2°C with illumination of 20 μmol photons $m^{-2}$ $s^{-1}$ by cool white 40W fluorescent tubes (Philips) After 15 days of incubation, when visible cyanobacterial growth appeared, colonies were isolated by repeated plating on 0.8%-agar with BG11 medium (Rippka et al. 1979), and successfully isolated colonies were transferred to liquid BG11 medium. Culture material from each strain (2 mL) was harvested during exponential growth

and centrifuged (10,000 $g$, 5 min). Genomic DNA was extracted from the cell pellet using the UltraClean DNA isolation kit (MoBio Laboratories, Solana Beach, CA, USA). 16S rRNA was amplified using primers PA (Edwards et al. 1989) and B23S (Lepère et al. 2000), PCR reaction and sequencing were performed as described in Casero et al. (2014).

### 2.6 DNA extraction procedures from natural samples, 16S rRNA gene libraries preparation and sequencing

Three individual rocks harbouring at least two of the three endolithic microhabitats were processed, which resulted in 11

samples, including technical replicates: cryptoendolithic (n=2), chasmoendolithic (n=6) and hypoendolithic (n=3). Colonization zone was scrapped and ground for DNA extraction. This DNA extraction was performed using 0.3 g of samples and the UltraClean DNA isolation kit (MoBio Laboratories, Solana Beach, CA, USA) with minor modifications. A two-step PCR strategy was used to prepare the sequencing libraries of endolithic microbial communities, as previously described (Robinson et al. 2015). DNA was amplified using primers 338F and 806R (V3-V4 hypervariable region) barcoded for

multiplexing; amplicons from 2 PCR reactions were pooled after the first step. Illumina paired-end sequencing (2 x 250bp) was performed using the MiSeq platform at the Johns Hopkins Genetic Resources Core Facility (GRCF).

### 2.7 Computational analysis

After demultiplexing and barcode removal, sequence reads with phred score<20 and length<100bp were discarded using sickle (Joshi and Fass, 2011), representing only 2% of the initial reads count. The Qiime package (v1.6.0) was used to further process

the sequences (Caporaso et al. 2010) and diversity metrics were calculated based on Operational Taxonomic Units (OTUs) at
the 0.03% cutoff against the Ribosomal Database Project (RDP) database release 11 (Cole et al. 2014). The resulting OTUs table was filtered of the rare OTUs (total abundance across all samples below 1%), representing 40% of the initial count (1511 OTUs).

## 2.8 Phylogenetic analysis

Sequences of 16S rRNA gene from Cyanobacterial OTUs that showed significant differences in their relative abundance between endolithic microhabitats and 16S rRNA gene sequences from cyanobacterial isolates, were aligned with sequences obtained from the NCBI GenBank using the Clustal W 1.4 software (Thompson et al. 1994). 16S rRNA gene sequences from GenBank were selected using the NCBI MegaBlast tool (http://blast.ncbi.nlm.nih.gov/ Blast.cgi, accessed 28.08.18). The final alignment length was 400 bp. Phylogenetic trees of each of the genes were constructed in MEGA 7.0 using the Maximum

Likelihood (ML) method (Kumar et al. 2016). The best-fitting evolutionary model, chosen following the BIC (Bayesian Inference Criterion) in MEGA 7.0, was the Kimura 2-parameter model (Kimura 1980) for 16S rRNA genes. 1000 bootstrap replicates were performed for all trees.

## 3. Results

We combined microclimate measurements, microscopy analyses and high throughput culture-independent molecular data to

identify the effect of micro-biogeography and the factors underlying the structure and composition of microbial assemblages of gypcrete endoliths from the hyper-arid Atacama Desert.

### 3.1 Sampling site

Gypcrete samples were collected from the Monturaqui area (MTQ), located in the Preandean Depression of the Atacama Desert (Fig. 1) on December 2015. Climate data recorded over a period of 22 months described a mean air temperature about 15°C,

with strong amplitude between minima and maxima (from -4.7°C to 49.3°C), average diurnal PAR ~ 1000 μmol photons m$^{-2}$ s$^{-1}$ with a maximum of 2553.7 μmol photons m-$^2$ s$^{-1}$, providing evidence for the extremely intense solar irradiance found in this region (Cordero et al. 2014). This area experiences extremely dry conditions, with an average air RH of about 20% with frequent lows of 1% and precipitations extremely scarce with mean annual values of 24.5 mm (Wierzchos et al. 2015). Gypcrete surface temperature examined with thermal infrared camera revealed a maximum temperature of 68°C.

### 3.2 Micromorphology of gypcrete

CT-Scan images provided a 3D spatial visualization of pore shapes and their distribution inside the gypcrete rock (Figs. 2). The pores revealed capillary-like micromorphology following a vertical orientation as is shown in both top and lateral views. Detailed 3D images pointed to the apparent absence of connectivity with the surface of most of the pores (Figs. 2). However, the presence of this connectivity cannot be discarded due to the limited resolution of the CT-Scan technique and the conditions



of acquisition. Moreover, CT-scan images of the gypcrete surface reveal undulated furrows due to the dissolution of gypsum after scarce rains (Video S1).

### 3.3 Endolithic microhabitats

Cross sections of the gypcrete rocks revealed the presence of three clearly differentiated microhabitats where a significant heterogeneity in micromorphology and structure was found (Figs. 3). The cryptoendolithic colonization zone was close to the
compact gypcrete surface layer (up to 5mm depth). Within cryptoendolithic microbial communities, two characteristic pigmented layers were distinguished. The observed orange colour belongs to microorganisms with high carotenoids content laid closest to the gypcrete surface. The green colour layer beneath the orange layer belong to microorganisms with chlorophyll and phycobiliproteins content. The presence of these pigments was previously reported by Wierzchos et al. (2015) and Vítek et al. (2016) (Fig. 3, A1). The chasmoendolithic colonization zone reached a deeper (up to 8mm depth) position in the substrate
and was directly connected to the surface (Fig. 3, B1). Finally, the hypoendolithic colonization zone, was located close to the compact bottom gypcrete crust, shaped like micro-caves (Fig. 3, C1).

Cyanobacteria were found in the cryptoendolithic habitat among lenticular gypcrete crystals, filling up vertically elongated pores, and aggregated around sepiolite nodules (Figs. 3, A2-A3), a clay mineral with high water retention capacity, previously identified in gypcrete by Wierzchos et al. (2015). SEM-BSE also revealed dense arrangements of cyanobacterial cells
embedded in concentric sheets of EPSs (Figs. 3, A3). By contrast, the microbial assemblages inhabiting the chasmoendolithic and hypoendolithic microhabitats were coating the walls of the cracks and caves previously described (Figs. 3, B2, B3, C2, C3). Detailed SEM-BSE (Figs. 3, A3-C3) and LM images (Figs. 3, A4-C4) of each microhabitat showed mainly Cyanobacteria with different size and morphology accompanied by heterotrophic bacteria.

### 3.4 Cyanobacterial isolates from endolithic microhabitats

A total of 12 cyanobacterial strains were isolated from the three different endolithic microhabitats (Table S1): 3 from cryptoendolithic, 3 from chasmoendolithic and 6 from hypoendolithic. The cyanobacterial strains were identified, following Komárek et al. (2014), as *Chroococcidiopsis* sp. (UAM800, UAM801, UAM802, UAM805, UAM808, UAM809, UAM810, UAM811), *Gloeocapsa* sp. (UAM803, UAM804) and *Gloeocapsopsis* sp. (UAM806, UAM807).

### 3.5 Structure and composition of endolithic communities

High throughput sequencing of 16S rRNA gene amplicons across 11 samples and 3 microhabitats resulted in a total of 385,440 V3-V4 SSU rDNA reads, with an average number of paired-end reads per sample of $35,040 \pm 6,288$ and an average length of $456 \pm 11$ bp. Diversity metrics, calculated from OTUs clustered at 97%, revealed no significant differences between microhabitats in terms of alpha-diversity (Table 1).

A total of 11 bacterial phyla with a relative abundance > 0.1% were found across all microhabitats. Of these only 7 had a
relative abundance over 1% of sequences across the different microhabitats (Figs. 4). Cyanobacteria, Proteobacteria,





Actinobacteria and Gemmatimonadetes were the most abundant phyla, representing 82%–83% of the total community (Fig. 4, A). Cyanobacteria dominated the communities inhabiting all endolithic microhabitats; in the cryptoendolithic and chasmoendolithic communities, Cyanobacteria did not exceed 40% of the sequences, while in the hypoendolithic community they reached a relative abundance of 60% (Fig. 4, A). Proteobacteria were the second most abundant phylum, contributing

~30% of the sequences in the cryptoendolithic and chasmoendolithic communities, and less than a half in the hypoendolithic community (13%). The relative abundance of Actinobacteria was even across all microhabitats, never exceeding 10% of the sequence reads. Gemmatimonadetes relative abundance showed differences across microhabitats representing 7%, 4.4% and 2.3% of sequences in the cryptoendolithic, chasmoendolithic and hypoendolithic communities, respectively (Fig. 4, A). Bacteroidetes and Thermi phyla also exhibited variation between the different endolithic communities, showing the higher

relative abundance in the hypoendolithic (8.2%) and cryptoendolithic (4.9%) microhabitats. Firmicutes and Planctomycetes were also found in all three microhabitats at very low relative abundance (0.003% and 0.002%). No archaeal OTUs were detected before or after the quality filtering of sequences during the samples processing.

The four main phyla constituted ~ 80% of OTUs, clustered at 97% identity, across all microhabitats, which was quite different from the distribution of sequence reads (Fig. 4, B). The three major phyla, Cyanobacteria, Proteobacteria and Actinobacteria,

has similar OTUs relative abundances across all three microhabitats (25%, 32% and 21% respectively). The greatest difference between the distribution of the relative abundance of sequences and that of OTUs is observed for Cyanobacteria in the hypoendolithic community.

Compared to other microhabitats this phylum showed the highest relative abundance in terms of sequences (60.4%) but the lowest in terms of OTUs (21.9%), thus revealing the high abundance of a very low number of cyanobacterial OTUs. Average

Bray Curtis distance confirmed that dissimilarity between microhabitats (CR-CH= 0.36, CR-HE= 0.44, CH-HE= 0.44) was higher than between replicates of the same microhabitat (CR=0.36, CH= 0.29, HE=0.32) Adonis and ANOSIM tests, performed with the 3 microhabitats categories (cryptoendolithic, chasmoendolithic and hypoendolithic), confirmed the statistical significance of the grouping ($R^2$ =0.38, $p$- value=0.014 and $R^2$=0.48, $p$-value=0.003 for adonis and ANOSIM, respectively).

**3.6 Cyanobacterial composition**

As the major component of the endolithic communities from the 3 described microhabitats, Cyanobacteria OTUs and isolates were studied in detail. A phylogenetic analysis of the 15 major cyanobacterial OTUs (relative abundance > 1%), and 12 isolates revealed 6 main clusters supported by high bootstrap values (Fig. 5).

Most of the OTUs (9 out of 15) and isolates (8 out of 12) were assigned to the *Chroococcidiopsis* genus and were distributed

in three clusters (I, III and V), each with representatives of *Chroococcidiopsis* isolates and clones' sequences from various deserts. Cluster I had the highest number of sequences from this study: six of the *Chroococcidiopsis* strains (UAM801, UAM810, UAM802, UAM809, UAM800, UAM808) and four of the cyanobacterial OTUs (OTU1, OTU497, OTU8, OTU112). This cluster also included two reference *Chroococcidiopsis* sp. sequences of soils from the Atacama Desert (Patzelt





et al. 2014). Cluster III included only *Chroococcidiopsis* isolate (UAM805), three OTUs sequences (OTU1772, OTU420 and

OTU4), and reference sequences of *Chroococcidiopsis* sp. strains isolated from quartz hypolithic communities from Chinese desert (Pointing et al. 2007) and from University Valley (Antarctica). The last *Chroococcidiopsis* sp. cluster, number V, had no sequence from isolates, two OTUs sequences (OTU7 and OTU98), sequences from cloning libraries from two deserts, Atacama and Jordan (Dong et al. 2007), and one *Chroococcidiopsis* sp. sequence from a Mediterranean biocrust (Muñoz-Martín et al. 2019).

Cluster II comprised cyanobacterial sequences belonging to the Nostocales order from the *Fischerella* and *Calothrix* genera to which OTU18 and OTU11 were respectively assigned. A total of 6 Cyanobacteria of this study were clustered with members of the *Gloeocapsa* and *Gloeocapsopsis* genera (order Chroococcales), four isolates (UAM806, UAM807, UAM804, UAM803) and two OTUs (OTU9, OTU854), forming cluster IV. Two reference sequences of *Synechococcus* together with OTU5 constituted Cluster VI.

Because of the low % identity of OTU2 with its closest relatives in the database (< 95%) (Table S2) and with our isolates sequences, it was not possible to provide an accurate taxonomical assignment for this OTU (Fig. 5). Hits were found between two of the isolates (*Chroococcidiopsis* UAM801 and *Gloeocapsopsis* UAM806) and two of the most abundant OTUs (OTU1 and OTU9, respectively).

Differential abundance analysis using DESeq2 test revealed that 9 out of 15 of the cyanobacterial OTUs were differentially

abundant in the three microhabitats (Fig. 6). Both OTUs 11 (*Calothrix* sp.) and OTU18 (*Fischerella* sp), phylogenetically assigned to the Nostocales order, were significantly more abundant in the chasmoendolithic community (3.8% and 1.5%, respectively) than in cryptoendolithic and hypoendolithic communities (< 0.4% for both OTUs) ($p$-value < 0.01). OTUs clustered with *Gloeocapsa* and *Gloeocapsopsis* (cluster IV), with *Synechococcus* (cluster VI), and with *Chroococcidiopsis* sp. from three clusters (I, III and V), showed significantly different abundances ($p$-value < 0.001) between the hypoendolithic

community and that of the two communities from the upper side of the substrate (cryptoendolithic and chasmoendolithic). OTU8 (*Chroococcidiopsis* sp.) was the only one displaying a higher abundance in the hypoendolithic community, while OTU9 (*Gloeocapsopsis* sp.), OTU5 (*Synechococcus* sp.), OTU854 (*Gloeocapsa* sp.) and OTUs 1772 and 7 (*Chroococcidiopsis* sp.) had a higher abundance in the cryptoendolithic and chasmoendolithic communities. The unassigned Cyanobacterial OTU2 was mostly found in the hypoendolithic community ($p$-value < 0.0001) with an average relative abundance of more than 39%,

of the total community while its relative abundance in the other two communities was ~ 0.4%.

## 4. Discussion

In this study, we characterized the microbial communities inhabiting gypcrete collected from the Monturaqui area (Preandean Depression), which is of particular interest due to its location in the hyper-arid zone of the Atacama Desert. While endolithic colonization of the gypsum crust and gypcrete in this area has previously been studied (Dong et al. 2007, DiRuggiero et al.

2013, Wierzchos et al. 2015, Meslier et al. 2018), this is the first work in which cryptoendolithic, chasmoendolithic and



hypoendolithic communities have been characterized separately. The novelty of this study lies in the consideration of two different EMCs inhabiting two endolithic microhabitats located in the upper part of the substrate, and in the description of the structure and composition of the hypoendolithic microhabitat and its endolithic community, located at the bottom part of the substrate. This work was based on a multidisciplinary approach to elucidate the relationship between microhabitat architecture

and community composition of EMCs hosted in these different endolithic microhabitats coexisting within the same piece of rock.

The Monturaqui region, located in the Preandean Depression of the Atacama Desert has been found to harbour two different substrates colonized by microbial communities, namely gypcrete (Wierzchos et al. 2015) and ignimbrite, a volcanic rock (Wierzchos et al. 2013). Both substrates showed endolithic colonization and a lack of epilithic colonization (rock surface

colonization). The absence of this second type of colonization in any substrates from the Monturaqui region may be explained by the extremely arid microclimate of this area, including low relative humidity, high fluctuation of air and surface temperature, extreme high solar irradiation and scarce precipitation (Wierzchos et al. 2015). Monturaqui has been described as a hyper-arid area, showing an aridity index of 0.0075 (Wierzchos et al. 2013), based on the ratio mean annual precipitation (P) and potential evapotranspiration rate (PET) (P/PET), up to one order of magnitude lower than the limit established by Nienow (2009) for

the classification of hyper-arid regions (0.05). Specific measurements of surface temperature for gypcrete revealed values of almost 70°C on its surface at zenith, thus approximating the upper temperature limit for photosynthesis of 74°C, under which thermophilic cyanobacteria in hot springs have been found to live (Castenholz et al., 2001). The combination of these environmental conditions has led to the avoidance of epilithic colonization in favour of endolithic colonization.

Potential endolithic habitability is tightly linked to the porosity of a lithic substrate because the distribution and size of pores are often directly related to the substrate's water retention capacity (Cámara et al. 2015; Herrera et al. 2009; Matthes et al.

2001; Omelon 2008; Pointing et al. 2009; Meslier et al. 2018). Porosity in gypcrete allows microbial communities to survive in different microhabitats, providing sufficient space for the communities, while receiving enough light and having enough water to grow. The porous network of gypcrete slows down water loss by rapid evaporation and helps its retention by capillary forces acting in small capillary-like shape pores. The inner architecture of gypcrete allows the habitability of three different

locations inside the substrate. The CT-Scan and SEM-BSE images from this work showed that all three types of microhabitats shared a vertical axis of morphology with vertical cracks constituting the chasmoendolithic (CH) microhabitat and capillary-like pores constitute the cryptoendolithic (CR) and hypoendolithic (HE) microhabitats. This capillary like pore architecture found in the CR microhabitat could be explained by the progressive substrate dissolution due to scarce rains and by the water retained and condensed within the micropores, as it occurs in halite endolithic microhabitats (Wierzchos et al. 2012a). The

observed HE microhabitat architecture consolidates the proposal of Wierzchos et al. (2015), in which the authors described the presence of a dense crust delimiting the bottom part of the HE microhabitat. This structure reveals different dissolving and crystallization processes of the gypsum following the water gravity flow and giving rise to the cave-shaped pores, thus providing this HE microhabitat with a hard permeable bottom gypsum layer.



The larger distance between the HE microhabitat and the top surface microhabitats CR and CH, might be thought as a limiting

factor for the development of HE communities, especially in terms of water availability. However, the location of the HE microhabitat at the bottom of the rock could reduce water losses due to evaporation processes. Thus, the micro-cave structures we observed in the HE microhabitat might retain liquid water for longer times, leading to cyanobacterial growth.

The structural characteristics of the crypto- and chasmoendolithic microhabitats, located at the top of the substrate, also allow access to water for the EMCs. Within the CR microhabitat the labyrinth of pores directly or indirectly connected to the surface

may act as cavities where water might be retained, condensed, and also be present in form of saturated water vapour (high RH) through the substrate, and be available to the microbial communities. Additionally, the presence of sepiolite inclusions improves water retention in those pores, as previously described (Wierzchos et al. 2015, Meslier et al. 2018), leading to lower rates of water loses by evaporation and gravitational forces. In contrast, the CH microhabitat provides direct access to rainfall liquid water for its community, via its fissure and cracks, while at the same time lowering water retention capacity by higher

evaporation rates and losing liquid water by percolation through the rock

Microbial communities inhabiting all three microhabitats were found in the form of large aggregates and were often embedded in an EPSs matrix. These characteristics are closely linked to survival strategies under harsh environmental conditions related to low water and nutrient availability (Billi 2009, Wright et al. 2005) . Since water is the most limiting factor for the development of microbial communities inhabiting endolithic microhabitats of gypcrete, it is the component on which adaptive

strategies are primarily focused. EPSs, because of their role in hydration and dehydration processes in lithobiontic communities from Antarctic deserts (de los Ríos et al 2007) and from the Atacama Desert (Dong et al. 2007; Wierzchos et al. 2011; Wierzchos et al. 2015; Crits-Christoph et al. 2016) are an essential adaptation strategy against hyper-aridity. The aggregates-like structure of these communities composed by cyanobacteria and other heterotrophic bacteria with a different physiological status also helps their survival against drought, since dead cells could provide physical protection against desiccation processes

(Postgate 1967; Roszak and Colwell 1987; Billi 2009; de los Ríos et al. 2004). In the case of the CR community, a special strategy against dryness was observed in this work, since microorganisms were located close to the sepiolite, as previously reported with respect to gypcrete endolithic communities (Wierzchos et al. 2015, Meslier et al. 2018). EPS and dead cells taking part in the aggregates can also act as a nutrient reservoir in such an oligotrophic environment as the endolithic microhabitats; since low amounts of water soluble ions were previously detected in the MTQ gypcrete (Meslier et al. 2018).

The absence of significant differences in diversity metrics between the three EMCs of gypcrete is in accordance with the diversity values of previously reported EMCs in the Atacama Desert (rev. in Casero et al. 2020)At a phylum level, the community was composed of three main dominant phyla, Cyanobacteria, Proteobacteria and Actinobacteria (Fig. 4) as in other EMCs of the Atacama Desert (Wierzchos et al. 2015, Meslier et al. 2018, Dong et al. 2007). However, a switch in the Proteobacteria and Actinobacteria relative abundances was found compared to gypcrete cryptoendolithic communities

previously described (Meslier et al. 2018). That difference is presumably associated with different DNA extraction methods and the inherent associated biases. While the three types of gypcrete microhabitats are exposed to the same climatic conditions,





we suggest that differences in micro-architectures resulted in drastically different sets of characteristics for water retention: CR counts on porous condensation and sepiolite, CH has an easier access to liquid water, and HE suffers less water loss.

While the communities from the 3 microhabitats had similar alpha diversity metrics, we found the composition of these
communities was statistically different, which is supported by the relative abundance of the main phyla, Cyanobacteria, Proteobacteria and Actinobacteria, across the microhabitats distributed differentially, exhibiting differences between the CR and CH communities as compared to the HE community, especially regarding cyanobacterial OTUs. This notable difference in the relative abundance of cyanobacteria could be related to the particular resources of the phototrophic community. The differential access to solar irradiance could explain the contrast between cyanobacterial proportions on both sides, at the top
(CR and CH) and bottom (HE) of the substrate. Thus, an update to the proposal by Wierzchos et al. (2018) is here suggested, in which a causal link is evoked to explain the higher abundances of phototrophs as opposed to heterotrophs in EMCs, which has been observed previously (Robinson et al. 2015, DiRugiero et al. 2013, Wierzchos et al. 2015, Meslier et al. 2018). According to that work, the scarcity of water was suggested to cause a lower metabolic activity in phototrophs, thus leading to a lower support of the heterotrophic community. However, in this scenario, light intensity should also be considered a crucial
factor in understanding the differences between the composition of top and bottom EMCs, since the HE community has a notably lower access to sun radiation. Thus, for EMCs communities based on phototrophic microorganisms, a limitation to one of those resources essential for photosynthesis would further lead to low rates of $CO_2$ fixation and, consequently, to a smaller heterotrophic community.

While we found multiple phylotypes of cyanobacteria among the gypcrete microhabitats, most of them belonged to the genus
*Chroococcidiopsis*. Several strains of this genus have previously been described in EMCs from both hot and cold deserts (Friedmann 1980) as a result of their capacity to cope with extreme environmental conditions (Billi et al. 2011; Verseux et al. 2017). Further supporting the different micro-environmental conditions and community composition between the top CR and CH habitats and the bottom HE habitat, was the discovery of an unclassified cyanobacterial OTU (UC-OTU, New Reference OTU2), which was almost exclusive to the HE microhabitat. Although the low percentage of sequence similarity did not allow
for an accurate taxonomical assignment, its closest relatives (~94% sequence identity for 450 nt of the 16S rRNA gene) were from habitats where light is the limiting factor for photosynthesis such as a pinnacle mat at 10 m depth from a sinkhole (Hamilton et al. 2017) and groundwater sample from a tectonically-formed cavern (Table S2). The unidentification of the UC-OTU highlights the importance of greater efforts in terms of isolation and characterization of cyanobacteria, especially from these environments.

The differential distribution of key members of these EMCs among microhabitats in the same lithic substrate and the same piece of rock, as their primary producers, reveals an "environmental filtering" process (Kraft et al. 2015). This concept focuses on the relationship between an organism and the environment, recognizing that not all organisms will be able to establish themselves successfully and persist in all abiotic conditions. Thus, in this scenario, the abiotic conditions linked to the architecture and location of the endolithic microhabitat would force the development of community assemblages highly
specialized to small scale differences, thereby exhibiting a microbiogeographical behaviour.





## 5. Conclusions

This study is the first to address differences between microbial communities inhabiting three differentiated endolithic microhabitats within the same lithic substrate. In this study, liquid water availability was confirmed to be a driver of community composition because the specific architectural features of each microhabitat facilitated water input and retention in different

ways. Water, light, and $CO_2$, are indispensable resources for photosynthetic activity. Thus, we support the cause and effect relationship where the restriction of these factors may affect the proportion of phototrophic and heterotrophic components in the EMC communities as proposed by previous works (Robinson et al. 2015, Wierzchos et al. 2018 and Meslier et al. 2018). The *Chroococcidiopsis* genus displayed a variety of strains distributed among all microhabitats, proving its high capacity to colonize effectively endolithic microhabitats under polyextreme conditions. Nevertheless, the presence of a singular

cyanobacterial OTU stresses the need for additional efforts in cyanobacterial characterization from these extreme environments.

Findings from this work reveal the importance of using an appropriate scale for the study of microbial communities. Indeed, we found that the microstructural and microarchitectural features of the endolithic habitats were key factors in determining the composition of endolithic microbial communities. Thus, this study suggests a cautious use of "macroenvironmental"

parameters in characterizing differences between endolithic communities from different deserts or substrates. Our results point to the need for a more thorough description of the micro-environmental conditions that directly exert an effect on microbial assemblages: light, water and $CO_2$. Therefore, once the relationship between factors affecting the absence and/or presence of certain taxa, the actual environmental filtering in these microhabitats could be described in more detail, it will be possible to draw on conclusions on the interactions and specific roles of the different members in the community.


## Data availability

All the sequencing data sets generated in this study have been submitted to the National Center for Biotechnology Information (NCBI) SRA database, and can be found under the BioProject ID PRJNA637482. Author contributions

## Author contributions

MCC and JW designed and performed the research. JW conceived the original project. MCC, JW and AQ wrote the manuscript; MCC, JW and CA performed the microscopy; TK contributed to CT-SCAN analysis; MCC, VMA and JDR contributed to the molecular data, analysis, and performed the sequencing. All authors contributed to editing and revising the manuscript and approved this version for submission.



## Acknowledgements

This study was supported by grant PGC2018-094076-B-I00 from MCIU/AEI (Spain) and FEDER (UE) to MCC and JW, by NSF grant DEB1556574 and NASA grant NNX15AP18G to JDR. The work of MCC was supported by grant BES 2014-069106 from the Spanish Ministry of Science and Innovation (MCINN). The MNCN-CSIC, Madrid, Spain is acknowledged for microscopy services.

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



**Table 1: Diversity estimates of microbial communities in the endolithic microhabitats of gypcrete.**

| Microhabitats | | Chao | OTU Richness | Shannon |
|---|---|---|---|---|
| Cryptoendolithic | Avg | 583.8 | 430 | 6.3 |
| | SD | 43.2 | 38 | 0.2 |
| Chasmoendolithic | Avg | 574.9 | 419 | 6.1 |
| | SD | 46.0 | 29 | 0.1 |
| Hypoendolithic | Avg | 564.9 | 409 | 4.6 |
| | SD | 31.7 | 32 | 1.0 |


**Figure 1: Sampling location in the Atacama Desert. Monturaqui area:** MTQ (black diamond). (© Google Earth, image providers: Ladsat /Copernicus)





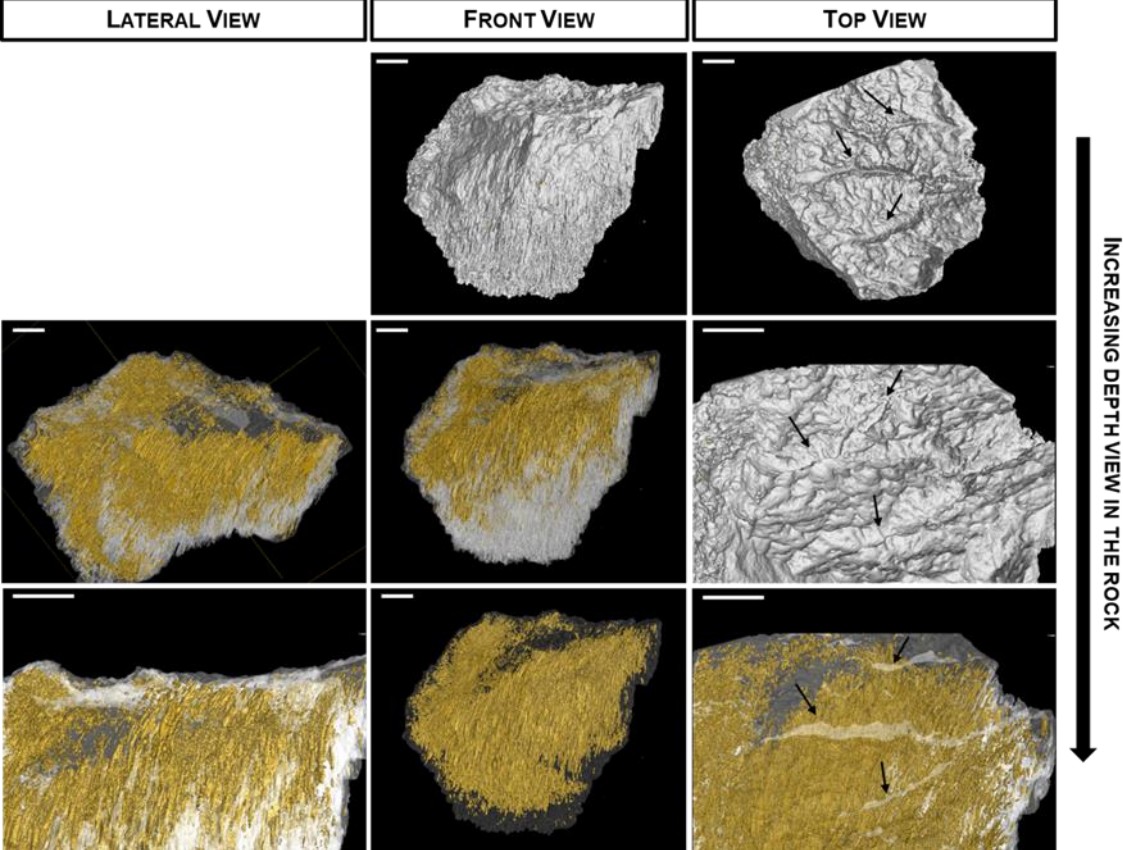

**Figure 2: CT-Scan images of a colonized piece of gypcrete.** 3D spatial distribution of pores (orange colour) and external view of the rock (grey colour) on lateral, front and top views of gypcrete. Porous micromorphology is capillary-shaped in vertical position due to gravity movement direction of water. Arrows in top view images point to the deepest cracks. Scale bar = 1cm.



**Figure 3: Endolithic colonization zones characterization.** Series A: Cryptoendolithic; Series B: Chasmoendolithic; Series C: Hypoendolithic. Series 1: Macro images of gypcrete cross-section of colonized zones; Series 2 and 3: SEM-BSE images of gypcrete cross-section of colonized zones; Series 4: LM-DIC images of scrapped cyanobacteria from gypcrete. **Series 1**: Black arrows indicate green and orange coloured endolithic colonization zones of 5 mm thick on A1 (CR), 8 mm thick on B1 beneath the surface (CH) and 5-9 mm thick on C1 above bottom gypsum crust (HE). **Series 2**. CR, CH and HE microhabitats with aggregates of endolithic microbial communities surrounded by green dotted lines, inside the pores of gypcrete: A2, under a white dense surface crust; B2, inside the cracks of gypcrete and C2, inside the micro caves of gypcrete at bottom of the rock. **Series 3** Green arrows point to aggregates of cyanobacteria among gypcrete (Gy) crystals (A3, B3), surrounding by sepiolite (Sp) nodules (A3) and on the gypcrete (Gy) walls (C3). **Series 4** aggregates of different morphotypes of cyanobacteria, shown by green, yellow and orange arrows and gypcrete crystals (Gy).





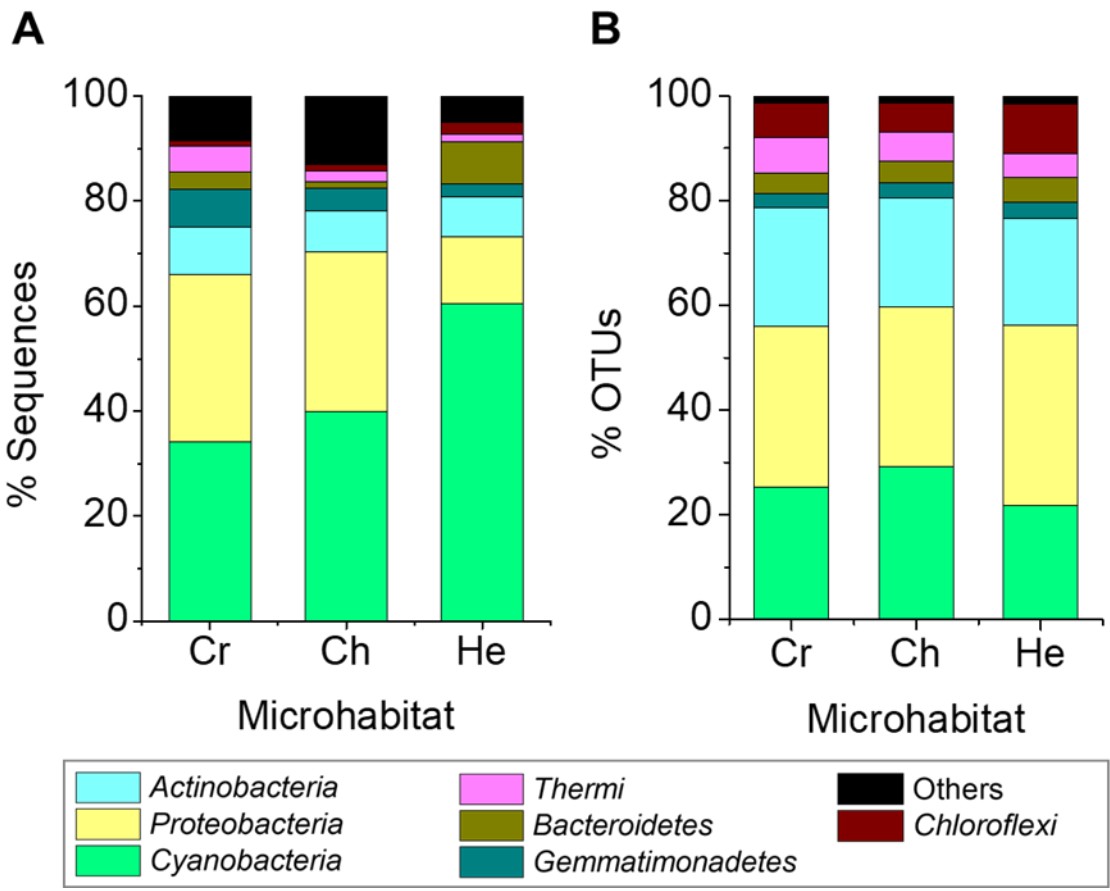

**Figure 4: Average relative abundance of sequence reads (A) and OTUs at the 97% clustering cut-off (B) of major bacterial phyla** (at least 1% across all samples) of microbial assemblages in the cryptoendolithic (Cr) chasmoendolithic (Ch) and hypoendolithic (He) microhabitats of gypcrete.



**Figure 5: Maximum likelihood tree based on partial 16S rRNA sequences of Cyanobacteria OTUs above 1% relative abundance and cyanobacterial strains isolated from the three gypcrete microhabitats**. Bold indicates sequences from this study. Scale bars indicates 5% sequence divergence.






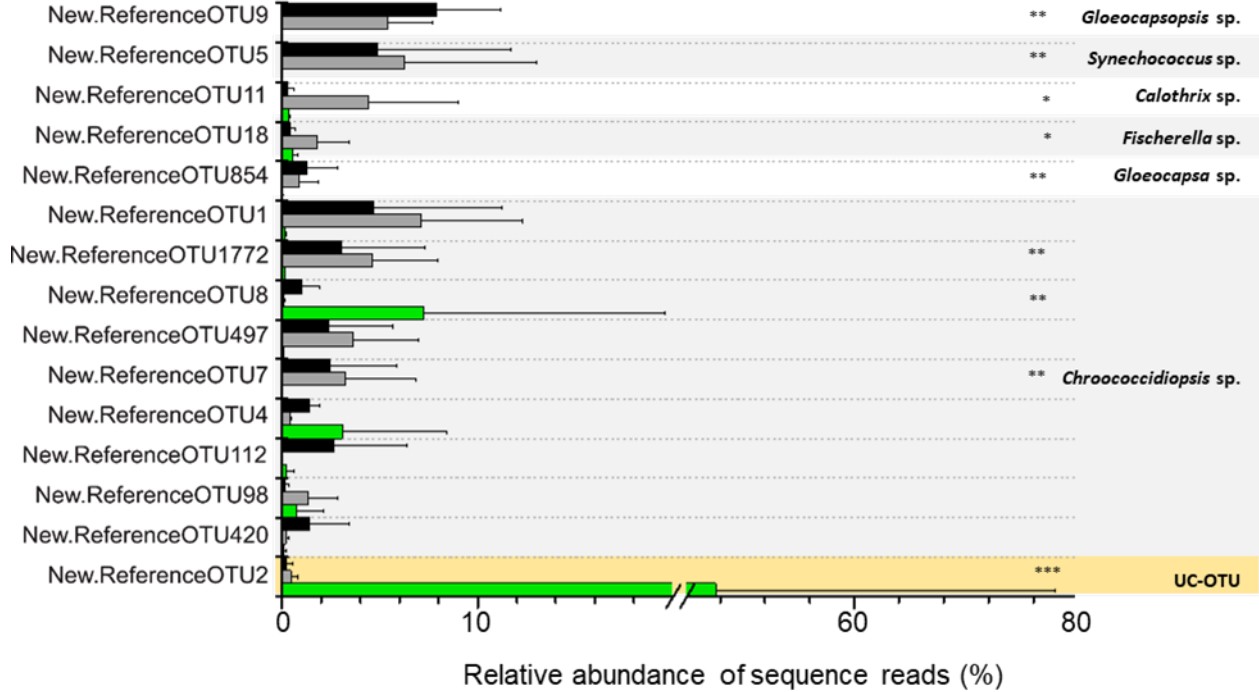

**Figure 6: Average relative abundance of cyanobacterial OTUs.** Differentially abundant cyanobacterial OTUs across the three microhabitats are represented by *(Diff-OTUs p-value <0.01 Ch / Cr-He), ** (Diff-OTUs *p*-value <0.001 He / Cr-Ch), ***(Diff-OTUs *p*-value <0.0001 He / Cr-Ch). UC-OTU (Unclassified Cyanobacterial OTU). Only sequences **> 1% relative abundances were used.**