# Peer review of "The composition of endolithic communities in gypcrete is determined by the specific microhabitat architecture"

_Biogeosciences, 2020_

## Referee Comment (RC1) · Anonymous Referee #1 · 20 Aug 2020

General comment. This work focuses on gypcrete as one of the lithic habitats found in the Atacama Desert in northern Chile. It provides new and important information on the architecture of this microhabitat and its impact on diversity and composition of endolithic lifeforms, complementing previous works carried out by this research team. The manuscript should be accepted for publication once the authors consider the following comments. Specific comments. 1. The last sentence on the Abstract (line 27) includes ". . . might be an essential factor in. . .". This rather ambiguous expression on the importance of microscopic features of endolithic habitats does not agree with the stronger terms found in the text on this major point (see Conclusions). I suggest that authors reconsider how to express their findings. 2. In reference to the UV radiation in Atacama by Cordero et al. (2018) at the Introduction (lines 49-50), authors should clarify that the highest measurements came from high altitude coastal and Andean sites and it does not apply to the whole territory since, as we know it today, UV radiation increases with the altitude. 3. I do not agree completely with the authors when they state (line 51) that life has found refuge in very specific endolithic (inside rocks) microhabitats. Microbial life has found not only endolithic habitats to cope with similar environmental conditions and several examples have reported life in other lithic locations at the coastal and hyperarid core of Atacama. Then, that sentence should be revisited. 4. On lines 60-61, authors emphasize the importance of the dominant genus Chroococcidiopsis leaving behind another cyanobacterial genus (Halothece), as part of more diverse lithic microbial communities than previously reported in earlier Atacama studies that include an accompanied microflora made of fungi and viruses and, supported by recent publications not properly credited in the manuscripts. 5. Authors indicate that they have coined "Microbiogeography" (line 73) as a new term. An important conceptual contribution whose scientific value will be validated by further studies in other lithic habitats, showing that gypcrete is not only a peculiar case. If the authors have information on this, they should stress it here to support the introduction of this new term and the international scientific community will have the opportunity to adopt it. Gypcrete samples came from a 3,000 m pre-Andean site, a quite different habitat when compared with others along Atacama but also, other endolithic substrates have other microscopic architectures depending upon their composition and crystal formation. I would like to have that author comments in their responses but also on the next revised manuscript. 6. Some parts of sections of Experimental procedures are brief, lack information and must be expanded or appropriate references should be added. Samples were taken during 2015; then, how storage time may have influenced the samples biodiversity? This a recurrent question and is important to know the authors position on this. Considering the microscale of the work, authors should clarify how they obtained samples from the three microhabitats involved in the study without "contamination". To learn about this strategy is of major importance if someone would replicate or apply

the protocols involved. This is finally the objective of the having a Material and Methods section in a paper. Cyanobacterial isolation was carried out from a bulk endolithic sample. Did the isolation strategy was independent of the inner location within the sampled rock? Did I understand correctly? Please, explain. DNA extraction was done with minor modifications. Well, modifications must be indicated. 7. Line 183. Alpha diversity differences were not found among the microhabitats. Then, do microhabitats affect colonization? Please, explain. Technical corrections. Line 22: "investigations": did you mean investigation? Line 107: add period after (Philips). Line 123: check for spaces at "score

---

## Author Comment (AC1) · 8 Sep 2020

We like to thank the reviewer for his/her comments, which helped us to improve our manuscript.

Specific comments:

1. The last sentence on the Abstract (line 27) includes ". . . might be an essential factor in. . .". This rather ambiguous expression on the importance of microscopic features of endolithic habitats does not agree with the stronger terms found in the text on this major point (see Conclusions). I suggest that authors reconsider how to express their

findings.

Authors: We agree with the reviewer comment and changed the sentence in the abstract, where now reads "plays an essential role in shaping the diversity and composition of endolithic microbial communities" (line 28).

2. In reference to the UV radiation in Atacama by Cordero et al. (2018) at the Introduction (lines 49-50), authors should clarify that the highest measurements came from high altitude coastal and Andean sites and it does not apply to the whole territory since, as we know it today, UV radiation increases with the altitude.

Authors: We agree with the reviewer and rephrase this section: "the highest surface ultraviolet radiation (UV), photosynthetic active radiation (PAR) and annual mean surface solar radiation (Cordero et al. 2018) in the Coastal Cordillera and Andean sites." (lines 51-52)

3. I do not agree completely with the authors when they state (line 51) that life has found refuge in very specific endolithic (inside rocks) microhabitats. Microbial life has found not only endolithic habitats to cope with similar environmental conditions and several examples have reported life in other lithic locations at the coastal and hyperarid core of Atacama. Then, that sentence should be revisited.

Authors: We agree with the reviewer that microbial life has not only been found in endolithic microhabitats but also in epilithic and hypoendolithic locations in the substrate. The sentence has been rephrased and now reads "In this inhospitable polyextreme desert, microbial life has been found in different lithic habitats, as epilithic (on rocks), hypolithic (under rocks) (Azua-Bustos et al., 2012) and endolithic (inside rocks) microhabitats (rev. by Wierzchos et al. 2018; Wierzchos et al. 2012b)." (lines 53-55).

4. On lines 60-61, authors emphasize the importance of the dominant genus Chroococcidiopsis leaving behind another cyanobacterial genus (Halothece), as part of more diverse lithic microbial communities than previously reported in earlier Atacama studies

that include an accompanied microflora made of fungi and viruses and, supported by recent publications not properly credited in the manuscripts.

Authors: Chroococcidiopsis genus is emphasized in the manuscript as the main member of most endolithic communities, especially in the Atacama Desert, and due to its demonstrated tolerance to diverse extreme environmental conditions. However, we agree with the reviewer that Chroococcidiopsis is not the only genus previously found in endolithic communities from this desert. Thus we rephrased that section as follows: " Molecular and microscopy characterization of these endolithic microbial communities shows that, overall, these communities are dominated by Cyanobacteria, mostly from the extremely resistant to ionizing radiation and desiccation Chroococcidiopsis genus (Meslier et al. 2018, Crits-Christoph et al. 2016, Billi et al. 2000, Cockel et al. 2005) as well as members from Gloeocapsa (Crits-Christoph et al. 2016) and Halothece (de los Ríos et al. 2010; Robinson et al. 2015; Uritskiy et al. 2019) genera" (lines 62-66).

5. Authors indicate that they have coined "Microbiogeography" (line 73) as a new term. An important conceptual contribution whose scientific value will be validated by further studies in other lithic habitats, showing that gypcrete is not only a peculiar case. If the authors have information on this, they should stress it here to support the introduction of this new term and the international scientific community will have the opportunity to adopt it. Gypcrete samples came from a 3,000 m pre-Andean site, a quite different habitat when compared with others along Atacama but also, other endolithic substrates have other microscopic architectures depending upon their composition and crystal formation. I would like to have that author comments in their responses but also on the next revised manuscript.

Authors: We are grateful to Referee #1 for him/her comments regarding the importance of the conceptual contribution of the term "microbiogeography". However, our results show, for the first time, that the differences in the architecture of a microhabitat, even within the same piece of lithic substrate, plays an essential role in shaping the diversity and composition of endolithic microbial communities. In this context, we are aware that

more "microbiogeographic" studies should be done with other endolithic habitats from the Atacama Desert and elsewhere. Thanks to Referee comment following sentence was added to the end of the Discussion section: . . ."However, we are aware that more "microbiogeographic" studies should be done with other endolithic microhabitats from the Atacama Desert and elsewhere showing that gypcrete is not only a peculiar case where differences in the architecture of a microhabitat play an essential role in shaping the diversity and composition of endolithic microbial communities". (lines 364-366)

6. Some parts of sections of Experimental procedures are brief, lack information and must be expanded or appropriate references should be added. Samples were taken during 2015; then, how storage time may have influenced the samples biodiversity? This a recurrent question and is important to know the authors position on this.

Authors: Sampling was performed in December 2015 and DNA extraction was done in March 2016. During that period, samples were kept in sterile bags and stored at room temperature, dry and dark conditions as explained in lines 80-81. We believe that those conditions do not facilitate the growth of microorganisms (lack of humidity and light – for phototrophs) and therefore minimize the effect that storage can have on the diversity observed after DNA extraction compared to that in the original sample.

RC- 6 (cont.) Considering the microscale of the work, authors should clarify how they obtained samples from the three microhabitats involved in the study without "contamination". To learn about this strategy is of major importance if someone would replicate or apply the protocols involved. This is finally the objective of the having a Material and Methods section in a paper.

Authors: We agree with the reviewer and explained in detail the followed procedure to avoid contamination between samples in 2.6 section (lines 121-126) that now reads: "Colonization zone was scrapped and ground for DNA extraction. To avoid contamination between samples from different microhabitats, the scraping of material was carried out in the following way: due to the possible proximity of both chasmoendolithic and

cryptoendolithic microhabitats, on the top of the rock, chasmoendolithic colonization zones more distant from cryptoendolithic colonization zones were selected. In addition, material from each of them was scraped avoiding the edges, so that material from different microhabitats could not be mixed. In the case of the samples coming from hypoendolithic samples, the distance from the other two microhabitats allowed their full scraping."

RC- 6 (cont.) Cyanobacterial isolation was carried out from a bulk endolithic sample. Did the isolation strategy was independent of the inner location within the sampled rock? Did I understand correctly? Please, explain.

Authors: Cyanobacterial isolation was performed from independent samples of each microhabitat, in the same manner as in the case of DNA extraction. Thus, scrapped material from each of the three different microhabitats in the study (cryptoendolithic, chasmoendolithic and hypoendolithic) was transferred to different BG11-agar plates. This procedure allowed us to classify the cyanobacterial isolates taking into account their original microhabitat as described in section 3.4 (lines 186-191) and Table S1. To clarify this procedure, the text has been rephrased: "Scrapped material from each endolithic colonization zone of gypcrete was transferred to different BG11 1.5%-agar plates (Purified agar, Condalab, Spain)" (lines 110-111).

RC- 6 (cont.) DNA extraction was done with minor modifications. Well, modifications must be indicated.

Authors: The description of the DNA isolation protocol has been updated including detailed modification. Now it reads: "This DNA extraction was performed using 0.3 g of samples and the UltraClean DNA isolation kit (MoBio Laboratories, Solana Beach, CA, USA) including a three-cycle step of freezing 0.3mL aliquots of sample suspended in buffer, breaking them down by using an adapted drill and melting in 60°C water bath, as described in Loza et al. (2013) and Becerra-Absalón et al. (2019)".

7. Line 183. Alpha diversity differences were not found among the microhabitats. Then,

do microhabitats affect colonization? Please, explain.

Authors: In this work, we did not focus on how microhabitat structure affects colonization but in how they affect the distribution of microorganisms related to their inner architecture. What we try to show is that, although the three gypcrete endolithic communities are not significantly different in terms of alpha-diversity, they are significantly different in terms of their composition and the distribution (relative abundances) of OTUs in each of these communities (Figure 4).

Technical corrections.

Line 22: "investigations": did you mean investigation? Authors: We agree with the reviewer; the word should be "investigation". Corrected in the manuscript.

Line 107: add period after (Philips). Authors: corrected

Line 123: check for spaces at "score Authors: checked and corrected

---

## Referee Comment (RC2) · Anonymous Referee #2 · 6 Oct 2020

Comments on: "The composition of endolithic communities in gypcrete is determined by the specific microhabitat architecture" by Casero, Meslier, DiRuggiero, Quesada, Ascaso, Kowaluk, & Wierzchos

General comments: In this paper the authors investigated the microbial associations present in three distinct microhabitats (cryptoendolithic, chasmoendolithic, and hypoendolithic) in samples of gypcrete from the polyextreme Atacama Desert using molecular techniques. They further try to tie differences in the associations to the architecture of the microhabitats. Although the fact that these three microhabitats harbor different associations Is not surprising, the work described is solid example of the use

of a multidisciplinary approach to the problem. The molecular sampling of different microhabitats from the same stone is also novel. Therefore, I am in favor of publishing the paper with only a few minor revisions. My only major concern is that they haven't really succeeded in separating the influence of the architecture of the microhabitat from the myriad of other environmental variables (light, moisture, temperature, chemistry, etc.) that may be influencing the development of the association.

Specific comments:

1. I am a little concerned about the reliance on molecular methods to characterize the associations. I understand such methods are necessary given the nature of the problem, but they sometimes overlook obvious features. In this particular case, the authors describe two differently pigmented layers in the cryptoendolithic habitat. The cause of this difference is not addressed. However, in Wierzchos et al. (2015), gypsum samples collected from a spot only a few miles away showed a similar pattern in the cryptoendolithic habitat. In this case the upper, orange-pigmented layer was dominated by eukaryotic algae. These do not appear in the present analysis. Are they absent? Or are they not picked up by the molecular methods used? There is also no discussion of whether the orange and green layers mentioned in the present paper represent different morphologies of the same association or different associations in the same microhabitat.

2. In the discussion on page 11 it is suggested that the water relations in the three microhabitats differ as a result of the architecture. A little elaboration here might help with the argument that architecture determines the association.

3. Have the authors considered the proposal by Friedmann and Sun concerning the relative proportions of mycobionts and phycobionts in lichens in response to temperature (Microbial Ecology 49:523-535) in the in relation to the authors hypothesis concerning the relative proportions of phototrophs and heterotrophs in extreme environments?

Technical comments:

1. Does the infrared camera used to measure surface temperature need to be calibrated to gypcrete in order to get an accurate temperature? Most systems need to take the emissivity of the surface material into account first.

2. I am not clear concerning the numbers of samples and replicates. It looks like three rocks were used. Two of these contained cryptoendoliths, three contained hypoendoliths, and all contained chasmoendoliths, and more than one chasmoendolithic association was sampled for each rock. Does this give enough statistical power for the analysis?

3. The CT scans by themselves are difficult to interpret (Figure 2).

4. I did not see any discussion of UAM811, which seems to hold a somewhat anomalous position in maximum likelihood tree (Figure 5).

Minor issues: 1. Azuá-Bustos et al. 2015 is missing from the references. Replaced by Azuá-Bustos & Gonzálex-Silva 2014?

2. Wierzchos et al 2012a and Wierzchos 2012b need to be differentiated in the references.

3. Cockell is misspelled in line 61

4. I prefer to put genus and phyla ahead of the names: "genus Chroococcidiopsis" instead of "Chroococcidiopsis genus".

5. Change "the limit established by Nienow (2009)" to "the established limit (Nienow 2009)" (line 264)—Nienow cited the limits but they were established previously.

6. Camara et al 2015 should be Camara et al 2014 (line 270)

7. Changes "consolidates" to "supports" (line 280)

8. instead of "unidentification" might be better to say "inability to identify." (line342)

9. Pointing references are run together. (line 485)

---

## Author Comment (AC2) · 15 Oct 2020

We like to thank the reviewer for his/her comments, which helped us to improve our manuscript.

Specific comments:

1. I am a little concerned about the reliance on molecular methods to characterize the associations. I understand such methods are necessary given the nature of the problem, but they sometimes overlook obvious features. In this particular case, the authors describe two differently pigmented layers in the cryptoendolithic habitat. The

cause of this difference is not addressed. However, in Wierzchos et al. (2015), gypsum samples collected from a spot only a few miles away showed a similar pattern in the cryptoendolithic habitat. In this case the upper, orange-pigmented layer was dominated by eukaryotic algae. These do not appear in the present analysis. Are they absent? Or are they not picked up by the molecular methods used? There is also no discussion of whether the orange and green layers mentioned in the present paper represent different morphologies of the same association or different associations in the same microhabitat.

Authors: We appreciate referee comment on the orange pigmented layer and its possible relation with the presence of eukaryotic algae. In this case algae were absent in all endolithic microhabitats of gypcrete in contrast with Wierzchos et al. (2015) samples and Meslier et al. (2018), where algae were not considered for analysis due to the OTU relative abundance filtering. We performed PCR of 18S rRNA gene in order to obtain eukaryotic sequences and obtained no amplification, also we did not find any OTU sequence belonging to algae chloroplast, which can occur when amplifying 16S rRNA gene from field samples in case algae are present. Also, microscopy observation of all three endolithic microhabitats in gypcrete did not reveal the presence of the algae. The following sentence has been added in the discussion to clarify the absence of algae in these gypcrete endolithic samples (lines 348-350): In contrast with results of Wierzchos et al. (2015) in gypcrete endolithic communities, no eukaryotic algae were found in neither microscopy nor molecular analyses, being Cyanobacteria the phototrophic phylum observed in all gypcrete endolithic microhabitats.

2. In the discussion on page 11 it is suggested that the water relations in the three microhabitats differ as a result of the architecture. A little elaboration here might help with the argument that architecture determines the association.

Authors: The discussion regarding the water relation with the specific features of each microhabitat is developed from line 298 to line 323. However, we agree with the referee that the text should be clarify. To indicate that this suggestion is related to what

is deeply discussed previously, that sentence now reads: Lines 330-333: While the three types of gypcrete microhabitats are exposed to the same climatic conditions, we suggest that differences in micro-architectures resulted in drastically different sets of characteristics for water retention discussed previously: CR counts on water capillary porous condensation and sepiolite water absorption properties, CH has an easier access to liquid water, and HE suffers less water loss.

3. Have the authors considered the proposal by Friedmann and Sun concerning the relative proportions of mycobionts and phycobionts in lichens in response to temperature (Microbial Ecology 49:523-535) in the in relation to the authors hypothesis concerning the relative proportions of phototrophs and heterotrophs in extreme environments?

Authors: We know the work of Friedmann and Sun (2005) in cryptoendolithic lichens. We understand their proposal about the ratio of photobiont and mycobiont in lichens. However, lichenic assemblages between algae and fungi are specific and no other organisms are involved in the symbiotic relationship. In our work we focused in the phototrophic members of the endolithic microbial community and that is why we only discussed about their influence with the ratio of all other members

Technical comments:

1. Does the infrared camera used to measure surface temperature need to be calibrated to gypcrete in order to get an accurate temperature? Most systems need to take the emissivity of the surface material into account first

Authors: Almost all infrared cameras need to be calibrated to measure surface temperature of any material. Gypcrete is almost composed by gypsum and for this material the emissivity values range from 0.8 to 0.95. However, we have introduced the value of 0.92. This value was obtained for gypcrete from sampling place equilibrated to temperature of 25 °C during 5 hours and the value of rock surface temperature detected by FLIR camera was adjusted to 25 °C by introducing adequate value (0.92) of emissivity. Following phrase was added to the text in M&M section (lines 93-94): Calibration of

the FLIR camera for measurements of gypcrete surface temperature was performed introducing the emissivity value of 0.92.

2. I am not clear concerning the numbers of samples and replicates. It looks like three rocks were used. Two of these contained cryptoendoliths, three contained hypoendoliths, and all contained chasmoendoliths, and more than one chasmoendolithic association was sampled for each rock. Does this give enough statistical power for the analysis?

Authors: Due to the problems associated with finding gypsum samples that had at least two of the three endolithic microhabitats under study, it was not possible to count on a large number of samples. However, the samples obtained, although few, were sufficient for the analyses carried out to have necessary statistical power.

3. The CT scans by themselves are difficult to interpret (Figure 2).

Authors: The Figure 2 information has been rewritten to clarify the interpretation. Now it reads: Figure 2: CT-Scan images of a colonized piece of gypcrete. 3D reconstruction of gypcrete sample with spatial distribution of pores (orange colour) and complete reconstructions of the scanned volume (grey colour) on lateral, front and top views of gypcrete. Porous micromorphology is capillary-shaped in vertical position due to gravity movement direction of water. Arrows in top view images point to the deepest cracks. Scale bar = 1cm. Also, 2D images of lateral and front view have been included in Supplementary Material to enable the correct interpretation of CT-Scan images.

4. I did not see any discussion of UAM811, which seems to hold a somewhat anomalous position in maximum likelihood tree (Figure 5)

Authors: Since the aim of this study is a multidisciplinary approach to the impact of microhabitat architecture in the diversity and composition of gypcrete endolithic microbial communities, we used several techniques and approaches obtaining diverse pieces of information. On the one hand, it allows us to combine all that information and helps us

to interpret it, giving a more complete picture of the endolithic communities of gypcrete. However, it makes an in-depth discussion of all the data obtained quite difficult, as is the case of isolated cyanobacteria, which would require a specific study on their own. Nevertheless, we agree with the referee that the phylogenetic position of UAM811 should be at least mentioned and taken into account in the discussion. Thus, that paragraph now reads: Lines 354-363: Further supporting the different micro-environmental conditions and community composition between the top CR and CH habitats and the bottom HE habitat, was the discovery of an unclassified cyanobacterial OTU (UC-OTU, New Reference OTU2), which was almost exclusive to the HE microhabitat and the phylogenetic distance of the hypoendolithic Chroococcidiopsis UAM811 strain with the different Chrooococcidiopsis clusters. Regarding the so called UC-OTU, although the low percentage of sequence similarity did not allow for an accurate taxonomical assignment, its closest relatives (∼94% sequence identity for 450 nt of the 16S rRNA gene) were from habitats where light is the limiting factor for photosynthesis such as a pinnacle mat at 10 m depth from a sinkhole (Hamilton et al. 2017) and groundwater sample from a tectonically-formed cavern (Table S2). Both observations, the inability to identify the UC-OTU and the phylogenetic position of the UAM811 strain, highlight the importance of greater efforts in terms of isolation and characterization of cyanobacteria, especially from these environments.

Minor issues:

1. Azuá-Bustos et al. 2015 is missing from the references. Replaced by Azuá-Bustos & Gonzálex-Silva 2014? Authors: Corrected (lines 404-405) 2. Wierzchos et al 2012a and Wierzchos 2012b need to be differentiated in the references. Authors: Corrected (lines 555, 563) 3. Cockell is misspelled in line 61 Authors: Corrected (line 64) 4. I prefer to put genus and phyla ahead of the names: "genus Chroococcidiopsis" instead of "Chroococcidiopsis genus". Authors: Modified genus (lines 63, 227, 374), genera (238, 240) and phyla (66, 68, 207) 5. Change "the limit established by Nienow (2009)" to "the established limit (Nienow 2009)" (line 264)âËŸAËĞ TNienow cited the

limits but they were established previously Authors: Changed (line 277) 6. Camara et al 2015 should be Camara et al 2014 (line 270) Authors: Corrected (line 284) 7. Changes "consolidates" to "supports" (line 280) Authors: Changed (line 293) 8. instead of "unidentification" might be better to say "inability to identify." (line342) Authors: Changed (lines 355-356) 9. Pointing references are run together. (line 485) Authors: Corrected (line 513)

Please also note the supplement to this comment:
https://bg.copernicus.org/preprints/bg-2020-245/bg-2020-245-AC2-supplement.pdf

―――――――――――――――

**Supplement:**

**SUPPLEMENTARY MATERIAL**

**Table S1: Cyanobacterial strains isolated from cryptoendolithic, chasmoendolithic and hypoendolithic microhabitats of gypcrete from MTQ.**

| Microhabitat | Strain code | Taxonomical Assignment |
| --- | --- | --- |
| **Cryptoendolithic** | UAM807 | *Gloeocapsopsis* sp. |
| | UAM808 | *Chroococcidiopsis* sp. |
| | UAM801 | *Chroococcidiopsis* sp. |
| **Chasmoendolithic** | UAM805 | *Chroococcidiopsis* sp. |
| | UAM806 | *Gloeocapsopsis* sp. |
| | UAM800 | *Chroococcidiopsis* sp. |
| **Hypoendolithic** | UAM802 | *Chroococcidiopsis* sp. |
| | UAM803 | *Gloeocapsa* sp. |
| | UAM804 | *Gloeocapsa* sp. |
| | UAM809 | *Chroococcidiopsis* sp. |
| | UAM810 | *Chroococcidiopsis* sp. |
| | UAM811 | *Chroococcidiopsis* sp. |

**Table S2: Taxonomical assignment of cyanobacterial OTUs by BLASTn to sequences belonging to uncultured and cultured material.**

| | CYANOBACTERIAL OTUs | | | | | | | |
|---|---|---|---|---|---|---|---|---|
| | Uncultured | | | | Cultured | | | |
| | BLASTn | Accession Number | Identity (%) | Environment | BLASTn | Accession Number | Identity (%) | Environment |
| OTU18 | Uncultured cyanobacterium clone 332-12 | KT453633 | 99 | Sublacustrine thermal vents Yellowstone Lake | *Chroococcidiopsis* sp. CC4 | DQ914866 | 99 | China quartz hypoliths |
| OTU11 | Uncultured cyanobacterium clone FWS-B15 | KC437357 | 100 | Hot Spring | *Calothrix* sp. NIES-3974 | AP018254 | 100 | |
| OTU854 | Uncultured *Gloeocapsa* sp. clone HL4SH30 | LN880050 | 97 | shoots of Haloxylon in high salinity | *Gloeocapsa* sp. PKUAC-GDTS1-13 | MG822744 | 97 | |
| OTU9 | Uncultured cyanobacterium clone Alchichica_AQ2_1_1C_10 | JN825312 | 99 | microbialites from Alchichica alkaline lake | *Gloeocapsa* sp. Ryu5-15d | LC325265 | 99 | blackened part of a surface of a building |
| OTU497 | Uncultured *Chroococcidiopsis* sp. clone ATA4-8-EC03 | KC311895 | 95 | soil Atacama Desert | *Chroococcidiopsis* sp. A789-2 | JF810071 | 94 | Antarctica: University Valle |
| OTU420 | Uncultured cyanobacterium clone IGW2-36 | KP238411 | 98 | volcanic rock ignimbrite, Atacama Desert, Lomas de Tilocalar | *Chroococcidiopsis* sp. RQEC | KY303728 | 97 | Hypolith quartz Taklimankan desert, Xingjiang |
| OTU1 | Uncultured *Chroococcidiopsis* sp. clone ATA4-8-EC03 | KC311895 | 98 | soil Atacama Desert | *Chroococcidiopsis* sp. CC1 | DQ914863 | 96 | quartz hypoliths China |
| OTU4 | Uncultured cyanobacterium clone IGD2-37 | KP238398 | 98 | volcanic rock ignimbrite, Atacama Desert, Lomas de Tilocalar | *Chroococcidiopsis* sp. A789-2 | JF810071 | 99 | Antarctica: University Valle |
| OTU1772 | Uncultured cyanobacterium clone IGW2-36 | KP238411 | 96 | volcanic rock ignimbrite, Atacama Desert, Lomas de Tilocalar | *Chroococcidiopsis* sp. RQEC | KY303728 | 96 | Hypolith quartz Taklimankan desert, Xingjiang |
| OTU98 | Uncultured cyanobacterium clone AY6_21 | FJ891051 | 99 | quartz, Yungay, Atacama Desert | *Chroococcidiopsis* sp. RQEC | KY303729 | 95 | Hypolith quartz Taklimankan desert, Xingjiang |
| OTU8 | Uncultured cyanobacterium clone AY6_17 | FJ891047 | 99 | quartz, Yungay, Atacama Desert | *Chroococcidiopsis* sp. CC1 | DQ914863 | 97 | quartz hypoliths China |
| OTU112 | Uncultured bacterium clone BJ201305-46 | KX507829 | 100 | rain water | *Chroococcidiopsis* sp. CC1 | DQ914863 | 97 | quartz hypoliths China |

| OTU2 | Uncultured bacterium clone LSS_Cyano_OTU5 | KP728185 | 95 | sinkhole lake | *Aphanocapsa muscicola* 5N-04 | FR798920 | 94 | fountain made of Sierra Elvira Stone, gray semi-dry patina on a water jet Spain:Granada, Generalife, Patio de la Sultana" |
|---|---|---|---|---|---|---|---|---|
| OTU5 | Uncultured cyanobacterium clone 3GA1-12_K89 | JX127189 | 99 | stone of castle wall Germany | *Synechococcus* sp. CIBNOR 42 | AY274622 | 99 | cyanobacterial bloom in the Urias estuary (Mazatlan, Sinaloa, Mexico) during a fish mortality event in spring 1999 |
| OTU7 | Uncultured bacterium clone Atacama-colB11 | EF071511 | 100 | Atacama Desert | *Chroococcidiopsis* sp. A789-2 | JF810071 | 94 | Antarctica: University Valley |

[Figure]

**Figure S2**. CT-Scan images of a colonized piece of gypcrete. 2D spatial distribution of pores (orange colour) and external view of the rock (grey colour) on lateral (left) and front (right) views of gypcrete. Porous micromorphology is capillary-shaped in vertical position due to gravity movement direction of water. Arrows in top view images point to the deepest cracks. Scale bar = 1cm.

**Video S1: CT-Scan film of a colonized piece of gypcrete. 3D spatial distribution of pores (orange colour) and external view of the rock (grey colour) on lateral, front and top views of gypcrete. Porous micromorphology is capillary-shaped in vertical position due to gravity movement direction of water.**

---

## Referee Comment (RC3) · Anonymous Referee #3 · 27 Oct 2020

This work focuses on documenting the composition and distribution of microorganisms within endolithic habitats in gypcrete in the Atacama Desert. It aims to test the hypothesis that specific microhabitat architecture influences microenvironmental conditions and therefore the relative abundance of different microorganisms within these habitats by using a combination of microscopy and molecular techniques.

The manuscript provides new and interesting information about microbial (relative) abundances within these habitats, and adequately explains how these habitats differ in terms of their physical architecture. For these reasons, I believe it is suitable for publication in Biogeosciences after considering the following comments:

[Figure]

Abstract: suggesting that the lithic substrate "might" be an essential factor does not in-still confidence in the results and conclusions, which contrasts with the term "confirms" in the Conclusions. In addition, the abstract should be more concise in describing the results of this work, not general observations of the results. For example, it currently points out that the hypoendolithic community was the least diverse and hosted unique taxa; explaining "why" here is important for the reader.

What is the significance of "Preandean Atacama Desert" within the context of this study?

Section 2.2: It is unclear how this climate data is directly relevant to the results of this manuscript. Other than thermal measurements, it does not appear to have been collected specifically for this work and so only needs to be mentioned in the Discussion.

Section 2.5: Title should include "DNA extraction procedures" to be consistent with Section 2.6.

Section 3: Results – Lines 139-141 are note necessary, nor is Section 3.1 with exception of gypcrete surface temperatures if they were measured for this study.

Section 3.3: Use present tense to describe observations, such as "...colonization zone is close..."

Section 4: What is the distance between the cryptoendolithic/chasmoendolithic habitats in the upper part of the substrate and the hypoendolithic habitat in the lower part of the substrate? Are they separated by millimetres? Centimetres?

Line 21 – "...a combination of microscopic investigations and..." Line 22 – "...the endolithic communities and their habitats at the microscale..." Line 23 – replace "lithic substrate" with "gypcrete" Line 39 – replace "noticeable" with "plausible" Line 39 – "...only by microorganisms that can survive and/or thrive under physical or geochemical extremes such as temperature... Line 43 – replace "stress" with "limitations" Line 45 – "...able to survive under such conditions" Line 46 – "The Atacama Desert...on Earth,

with scarce precipitation events…and extremely low mean annual relative humidity" Line 54 – replace "living inside the rock but close to the bottom" with "living on the underside of the rock" Line 56 – "…PAR radiation levels…" Line 67 – "…architecture on the diversity…" Line 71 – "…the microscale dimensions…" Line 71 – How do you define "peculiar"? Line 72 – Can you provide a definition for the term "microbiogeography"? Line 75 – "The area experiences…" Line 78 – "…we sampled gypcrete…" Line 80 – dry and dark environment – a lab drawer? Line 92 – "Light microscopy (LM) was used to examine cell aggregates…" Line 93 – "…on cyanobacterial isolates cultured from…" Line 97 – "…were run on pieces…" Line 101 – "…to reduce beam hardening." Line 102 – "..performed using VG Studio Max Version 2.2 software." Line 105 – "Biological material removed from gypcrete…" Line 105 – Was it BG11+N or BG11-N? Line 107 – Include a period after "Philips" Line 107 – "After incubation for 15 days, visible…" Line 116 – "Colonization zones were scraped…" (not scrapped) Line 130 – "Sequences of 16S rRNA genes…" Line 151 – Replace "visualization" with "representation" Line 152 – Replace "following" with "exhibit" Line 155 – Can you better describe "undulated furrows"? Line 160 – How did you differentiate pigmentation in layers? Light microscopy? Line 161 – "…with high carotenoid content closest to…" Line 219 – "…included only one…" Line 220 – Which Chinese desert are you referring to? Line 221 – Please provide a reference for University Valley Line 222 – "…no sequences from isolates…" Line 230 – "…with our isolate sequences…" Line 234 – Can you think of another way to say "differentially abundant"? Line 235 – "Both OTU11…" Line 255 – The last sentence of this paragraph is not necessary. Line 259 – "Both substrates show…" Line 263 – "…based on the ratio of mean…" Lines 265-267 – I do not know if you can compare temperatures on terrestrial rock surfaces with those in hot springs as an approximation for the upper temperature limit for photosynthesis. Can you estimate the temperatures within the endolithic habitats? Line 273 – "…water to metabolise and grow." Line 273 – "…gypcrete restricts water loss" Line 280 – I don't think "consolidates" is the correct word in this sentence Line 282 – "following water gravity flow" is unclear Line 295 – end the sentence with a period Line 297 – "EPSs"

should be "EPS" Line 300 – see line 297 Line 302 – "The aggregate-like structure..."
Line 303 – "...and heterotrophic bacteria also helps..." Line 308 – "...in such an olig-
otrophic environment." Line 330 – can you provide a reference that would support the
statement that light intensity should be considered a crucial factor in understanding
differences in community composition between top and bottom habitats? Line 342 –
replace "unidentification" with "lack of positive identification" Line 353 – Replace "con-
firmed" with "hypothesized"; I would not say that this work confirms that liquid water
availability is a driver of community composition, as no experimental evidence was
provided in the manuscript to substantiate this claim. A more convincing argument for
how microenvironmental conditions determines microbial distribution would strengthen
the manuscript. Line 369 – "...draw conclusions..."

Figure 1 – Latitude and longitude markers should be included in the study are map Fig-
ure 3 – It would be helpful to point out what samples are polished blocks/thin sections
vs whole mounts for SEM work.

---

## Author Comment (AC3) · 7 Nov 2020

We like to thank the reviewer for his/her comments, which helped us to improve our manuscript.

1. Abstract: suggesting that the lithic substrate "might" be an essential factor does not instill confidence in the results and conclusions, which contrasts with the term "confirms" in the Conclusions. In addition, the abstract should be more concise in describing the results of this work, not general observations of the results. For example, it currently points out that the hypoendolithic community was the least diverse and hosted unique taxa; explaining "why" here is important for the reader.

[Figure]

Authors: We agree with Reviewer 3 regarding of the use of the term "might" in the abstract. We changed the text and now reads: "These results show, for the first time, that the differences in the architecture of a microhabitat, even within the same piece of lithic substrate, plays an essential role in shaping the diversity and composition of endolithic microbial communities". Also, to better describe the results of the work, as suggested by the reviewer, we added "as a result of a lower access to sun radiation" after that sentence.

2. What is the significance of "Preandean Atacama Desert" within the context of this study?

Authors: Due to the huge extension of the Atacama Desert, the term Preandean is used in this manuscript to better localize the region of the sampling site, which also determines the climatic conditions of the area of study. The Preandean region of the Atacama Desert has been well defined in several Chapters of the book: "Microbial Ecosystems in Central Andes Extreme Environments.Biofilms, Microbial Mats, Micro-bialites and Endoevaporites" edited by M.E. Farías, Springer Nature Switzerland AG, 2020. In this book the Part II is referred to Preandean and Andean Atacama Desert: Life at Limits with Chapter 3: The Desert Polyextreme Environment and Endolithic Habitats by Jacek Wierzchos, Carmen Ascaso, Octavio Artieda, and María Cristina Casero and Chapter 4: Preandean Atacama Desert Endolithic Microbiology by María Cristina Casero, Victoria Meslier, Jacek Wierzchos, and Jocelyne DiRuggiero. We are grateful to Referee #3 for this comment and bibliographic reference of elsewhere mentioned book (chapter 4) was introduced to the manuscript were Preandean region of the Atacama Desert was mentioned.

3. Section 2.2: It is unclear how this climate data is directly relevant to the results of this manuscript. Other than thermal measurements, it does not appear to have been collected specifically for this work and so only needs to be mentioned in the Discussion

Authors: Our paper is focused on microbial ecology of very singular endolithic microbial

communities within gypcrete rocks in an extreme environment. Indeed, microclimate parameters such as air temperature and RH over a period of 22 months in the sampling place was only described once by Wierzchos et al. (2015). We consider these data of interest to the readers, such as microbial ecologists of extreme environments. Moreover, we have measured colonized gypcrete surface temperature revealing maximum of 68°C, which is very high temperature, even for desiccated endolithic microbial communities. Also this data was considered by us as of interest to the readers. We preferred to introduce detailed values of climatic and thermal measurements data in description of Sampling Site section (Results) as also these data were discussed in Discussion section.

4. Section 2.5: Title should include "DNA extraction procedures" to be consistent with Section 2.6.

Authors: Included in the title.

5. Section 3: Results – Lines 139-141 are note necessary, nor is Section 3.1 with exception of gypcrete surface temperatures if they were measured for this study.

Authors: Please, consider as correct our response as in point 3.

6. Section 3.3: Use present tense to describe observations, such as ". . . colonization zone is close. . . " Authors: We agree with Reviewer 3 and so we changed observations to present tense.

7. Section 4: What is the distance between the cryptoendolithic/chasmoendolithic habitats in the upper part of the substrate and the hypoendolithic habitat in the lower part of the substrate? Are they separated by millimetres? Centimetres?

Authors: Cryptoendolithic/chasmoendolithic microhabitats and hypoendolithic microhabitat are separated by centimetres ($\sim$ 4 cm).

Line 21 – ". . .a combination of microscopic investigations and. . ." Line 22 – ". . .the endolithic communities and their habitats at the microscale. . ." Line 23 – replace "lithic

substrate" with "gypcrete"

Authors: It now reads: "A combination of microscopy investigation and high-throughput sequencing approaches were used to characterize the endolithic communities and their habitats at the microscale within the same piece of gypcrete"

Line 39 – replace "noticeable" with "plausible" Line 39 – "...only by microorganisms that can survive and/or thrive under physical or geochemical extremes such as temperature..."

Authors: It now reads: "Regarding the second half of the statement (but, the environment selects) extreme environments present some of the most plausible scenarios since they are inhabited only by microorganisms that can survive and/or thrive in their respective physical or geochemical extremes such as temperature, solar radiation, pressure, desiccation, pH"

Line 43 – replace "stress" with "limitations":

Authors: Replaced

Line 45 – "...able to survive under such conditions"

Authors: accepted and changed

Line 46 – "The Atacama Desert...on Earth, with scarce precipitation events...and extremely low mean annual relative humidity"

Authors: accepted and changed

Line 54 – replace "living inside the rock but close to the bottom" with "living on the underside of the rock"

Authors: We agree with Reviewer 3 that the expression used is not clear. However we can not change it by the expression "underside of the rock" since it could be understood as the hypolithic colonization. Thus, we changed it and now it reads "hypoendolithic

(living inside pores in the bottom part of the rock)".

Line 56 – ". . .PAR radiation levels. . . "

Authors: accepted and changed

Line 67 – "...architecture on the diversity..."

Authors: accepted and changed

Line 71 – "...the microscale dimensions..."

Authors: accepted and changed

Line 71 – How do you define "peculiar"?

Authors: The term "peculiar" in this context is ambiguous so that we decided to change it by a term that better describes what we want to say. Now it reads: "The microscale dimensions and differential diversity distribution in this unique environment has led us to coin the new term "microbiogeography"

Line 72 – Can you provide a definition for the term "microbiogeography"?

Authors: Biogeography is a known term. However, our investigation describes for the first time the composition of endolithic communities at microscale, namely within few cm3 of lithic substrate. For this reason, we introduced in our manuscript the term "microbiogeography", since the composition of microbial communities is changing even at microscale, what was demonstrated in our work.

Line 75 – "The area experiences. . ."

Authors: accepted and changed

Line 78 – ". . .we sampled gypcrete. . ."

Authors: accepted and changed

Line 80 – dry and dark environment – a lab drawer?

[Figure]

Authors: Thank to Reviewer . #3 for this indication. Exactly, samples were stored in a lab drawer. However, to clarify the description we changed the sentence, that now reads: "...dry and dark conditions...".

Line 92 – "Light microscopy (LM) was used to examine cell aggregates..." Line 93 – "...on cyanobacterial isolates cultured from..."

Authors: accepted and changed

Line 97 – "...were run on pieces..."

Authors: accepted and changed

Line 101 – "...to reduce beam hardening."

Authors: accepted and changed

Line 102 – "..performed using VG Studio Max Version 2.2 software."

Authors: accepted and changed

Line 105 – "Biological material removed from gypcrete..."

Authors: accepted and changed

Line 105 – Was it BG11+N or BG11-N?

Authors: The culture medium used was BG11 with nitrogen (NaNO3). When BG11 culture medium contains no NaNO3, it would be called BG110. (sensu Rippka et al. 1979)

Line 107 – Include a period after "Philips"

Authors: accepted and changed

Line 107 – "After incubation for 15 days, visible..."

Authors: Accepted and changed. It now reads: "After incubation for 15 days, visible

cyanobacterial growth appeared. Colonies were isolated by repeated plating on 0.8%-agar with BG11 medium (Rippka et al. 1979), and successfully isolated colonies were transferred to liquid BG11 medium"

Line 116 – "Colonization zones were scraped. . ." (not scrapped)

Authors: accepted and changed

Line 130 – "Sequences of 16S rRNA genes. . ."

Authors: accepted and changed

Line 151 – Replace "visualization" with "representation"

Authors: accepted and changed

Line 152 – Replace "following" with "exhibit"

Authors: accepted and changed

Line 155 – Can you better describe "undulated furrows"?

Authors: We agree with Reviewer 3 that the description should be clarify. After consulting the literature, a better term for the observed dissolution features over gypcrete was found. Thus, we changed it by "microrills weathering features (DiRuggiero et al. 2013)"

Line 160 – How did you differentiate pigmentation in layers? Light microscopy?

Authors: The observation of different pigmentation layer was performed by stereoscopic microscopy

Line 161 – ". . .with high carotenoid content closest to. . ."

Authors: accepted and changed

Line 219 – ". . .included only one. . ."

Authors: accepted and changed

Line 220 – Which Chinese desert are you referring to?

Authors: After revising the literature corresponding to the cited sequence and GenBank database, it is not possible to assign that Chroococcidiopsis sequence to a specific location of those studied by Pointing et al. (2007): Qaidam Basin, Turpan Depression and Taklimakan Desert. However, to clarify the climatic conditions of the studied deserts by those authors, the sentence now reads: "Chroococcidiopsis sp. strains isolated from quartz hypolithic communities from hyperarid Chinese deserts (Pointing et al. 2007)"

Line 221 – Please provide a reference for University Valley

Authors: Included. Cumbers, J. and Rothschild, L. J.: Salt tolerance and polyphyly in the cyanobacterium Chroococcidiopsis (Pleurocapsales)., J. Phycol., 50(3), 472–482, doi:10.1111/jpy.12169, 2014.

Line 222 – "...no sequences from isolates..."

Authors: accepted and changed

Line 230 – "...with our isolate sequences..."

Authors: accepted and changed

Line 234 – Can you think of another way to say "differentially abundant"?

Authors: "Differentially abundant taxa" is a common term in this type of studies to define those taxa/features whose different abundance across samples is statistically significant. Examples: Taye et al. (2020) https://doi.org/10.3389/fmicb.2019.03007 Shatzkes et al. (2017) https://doi.org/10.1038/srep43483 Jiang et al. (2017) https://doi.org/10.1128/mSystems.00092-17

Thus, we consider that the term is correctly used in this study.

Line 235 – "Both OTU11..."

Authors: accepted and changed

Line 255 – The last sentence of this paragraph is not necessary.

Authors: We removed that sentence

Line 259 – "Both substrates show…"

Authors: accepted and changed

Line 263 – "…based on the ratio of mean…"

Authors: accepted and changed

Lines 265-267 – I do not know if you can compare temperatures on terrestrial rock surfaces with those in hot springs as an approximation for the upper temperature limit for photosynthesis. Can you estimate the temperatures within the endolithic habitats?

Authors: We agree with the reviewer that the comparison of rock surface temperatures with those in hot springs is inaccurate and that its relation to the temperature in the endolithic microhabitats should be mentioned. Thus, we revised that paragraph and now it reads:

"Specific measurements of surface temperature on gypcrete revealed values of close to 70°C. This value was detected at zenith, when microbial communities are desiccated and metabolically inactive (Cockell et al. 2008). The temperature within the endolithic habitats is expected to be close to that in the rock surface as shown by Wierzchos et al. (2012a) for halite endoliths."

Line 273 – ": : :water to metabolise and grow." Line 273 – ": : :gypcrete restricts water loss"

Authors: accepted and changed

Line 280 – I don't think "consolidates" is the correct word in this sentence

Authors: It now reads "supports the proposal of Wierzchos et al. (2015)"

Line 282 – "following water gravity flow" is unclear

Authors: We agree with Reviewer 3 and this phrase was corrected as follow: ..."
This structure reveals different dissolving and crystallization processes of the gypsum
following the water displacement from the surface to the bottom of the rock (gravity
flow). This water gravity flow giving rise to the cave-shaped pores, thus providing this
HE microhabitat with a hard permeable bottom gypsum layer" ...

Line 295 – end the sentence with a period

Authors: accepted.

Line 297 – "EPSs" should be "EPS" Line 300 – see line 297

Authors: accepted and changed

Line 302 – "The aggregate-like structure..."

Authors: accepted and changed

Line 303 – "...and heterotrophic bacteria also helps..."

Authors: accepted and changed

Line 308 – "...in such an oligotrophic environment."

Authors: accepted and changed

Line 330 – can you provide a reference that would support the statement that light in-
tensity should be considered a crucial factor in understanding differences in community
composition between top and bottom habitats?

Authors: Recently the light intensity, as driving factor of spatial heterogeneity within
halite endolithic microbial communities was reported by Uritskiy et al. (2020). This
phrase and ref. was introduced to the manuscript text.

Line 342 – replace "unidentification" with "lack of positive identification"

Authors: Changed. It now reads "the inability to identify the UC-OTU"

**BGD**

Line 353 – Replace "confirmed" with "hypothesized"; I would not say that this work confirms that liquid water availability is a driver of community composition, as no experimental evidence was provided in the manuscript to substantiate this claim. A more convincing argument for how microenvironmental conditions determines microbial distribution would strengthen the manuscript.

Authors: We agree with Reviewer 3 that confirmed is not an appropriate word to define the findings included in this work. Thus, the term it was changed and now it reads "In this study, liquid water availability was proposed to be a driver of community composition..."

Line 369 – "...draw conclusions..."

Authors: accepted and changed.

Figure 1 – Latitude and longitude markers should be included in the study are map

Authors: GPS coordinates are already included in site description and sampling section

Figure 3 – It would be helpful to point out what samples are polished blocks/thin sections vs whole mounts for SEM work.

Authors: no thin sections were included in Figure 3. All samples are polished blocks

---

## Author Comment (AC4) · 24 Nov 2020

We are grateful for Ref. # 2 and your comments focused on microhabitat's architecture and its relations with environmental factors in that very microhabitat. It was several years ago when we discovered the presence of endolithic microorganisms within several substrates in the hyperarid Atacama Desert as the last refugees of life in these harsh environmental conditions (review in Wierzchos et al. 2018). The attached Table summarizes the presence of dominant microorganisms within three different endolithic habitats (crypto-, chasmo- and hypoendolithic) in only one Ca-sulfate lithic substrates.

Hence the endolithic habitat could be the same, their dwelling microbial communities'

composition could be very different. It means that the denomination and nature of endolithic habitat is not a driver of microbial structure. If we compare the Ca-sulfate substrates from different climatic regimes of the Atacama Desert it is obvious that indeed these characteristics of climate regimes (T, RH, rainfall, dewfall, etc.) must have the main influence on the endolithic microbial communities' composition. Definitely, for this reason, our study was performed within the same external climate regime and more: within the same piece of gypcrete with three well defined endolithic microhabitats. This was a challenging question: how are the structure and composition of endolithic colonization within the same climatic regimen and the same piece of the rock? Our work answers that there are certain differences in endolithic microbial structure among crypto-, chasmo- and hypoendolithic habitats. We would like to again underline that the external climatic regime was absolutely the same for studied rock pieces and one could expect the same or very similar microbial structure colonization within all three endolithic habitats. However, our results have shown that indeed the structure of these microbial colonization's is different among endolithic habitats. How is a driver of these differences? Of course different microclimatic regimes at the micro-scale within different endolithic microhabitats. Obviously, it is impossible to measure microclimate parameters such as T, RH, dewfall, gravity water flow, water nanopore condensation, evaporation rates, solar irradiance, heat irradiance, etc. within endolithic microhabitats. However, it is and it was possible to describe and characterize the "architecture" of these microhabitats (this work and references in a review in Wierzchos et al., 2018). The term rock architecture was for the first time introduced by Wierzchos et al. (2015) as follows:

. . ."4.6. Architecture of a lithic substrate As observed with microscopy techniques, internal structural elements such as porosity, pore-size distribution, presence of large pores and cavities, light transparency and light scattering properties, dissolution and crystallization features, and sepiolite nodules distribution varied significantly within various location of the gypsum substrate. These all together structural, physical, chemical and mineral elements give rise to a new understanding of the features and functions

relevant to the rock bioreceptive characteristics. We suggest using the term of "rock architecture" instead of "rock structure" to emphasize the functional role of the rock interior. As such, this new concept of the architecture of a lithic substrate encompasses the internal structures of a rock with all mentioned elements that are essential as a habitat for microbial life. It is about perceiving the rock interior from the existence of porous spaces of different sizes and shapes, interconnected or not; the solid structures that divide and support these spaces, and the minerals and salts that can be transformed. All these components and elements are interrelated and influence one another, thus fulfilling a requisite: they might shape a suitable architecture to hold microbial life. The architecture of habitable rocks provides resources (water, light and nutrients, above all) and guarantees effective protection from excessive evapotranspiration, thus assuring efficient gas exchange and provides long-time stable fabric. Considering the architecture of a rock can provide an integrated view of its potential habitability for endolithic microbial communities. All porous rocks have a structure, yet very few show such a suitable architecture for endolithic microbial colonization, even under extreme environmental conditions, as the Atacama's gypsum do.". . .

Following this definition, we can distinguish different architecture of the substrate within different endolithic microhabitats, and indeed these differences will induce different microenvironmental characteristics on the microscale. And these microenvironmental characteristics shaping the different microbial structures within different endolithic microhabitats what was shown in our paper. As so, we do not pretend to separate the influence of the architecture of the microhabitat from the myriad of other environmental variables. Quite opposite. We consider that indeed distinct differences in the microarchitecture of the microhabitats have an influence on environmental variables at the microscale and shape microbial colonization structure. We consider that the endolithic communities are determined by endolithic microhabitat architecture and not by the endolithic microhabitat type (crypto-, chasmo- and hypoendo) (see Table 1). However, we agree that a much more precise conceptual definition of the above-mentioned relationships is needed and appropriate corrections were introduced in the text of the

manuscript as follows:

Introduction section:

The concept of rock architecture was introduced by Wierzchos et al. (2015) for colonized gypcrete substrate and encompasses the internal structures of rock with all elements that are essential for microbial life. Microhabitat architecture allows perceiving the rock interior from the existence of porous spaces of different sizes and also the solid structures that divide and support these spaces. All these components and elements are interrelated and influence one another, thus fulfilling a requisite: they might shape a suitable architecture to hold microbial life.

Discussion section:

Our work answers that there are certain differences in endolithic microbial communities' structure among crypto-, chasmo- and hypoendolithic habitats. Considering that the external climatic regime was the same for studied pieces of rock, our results have shown that the structure of these microbial communities was different among endolithic habitats. Following the definition of microhabitat architecture by Wierzchos et al. (2015) we can distinguish different architecture of the substrate within different endolithic microhabitats. In this context, our work suggests that distinct features of microhabitat architecture that have an influence on microenvironmental variables at the microscale would shape microbial communities' structure.

—————————————————————

**Table 1.** Dominant microorganisms within three different endolithic habitats. *Endolithic algae-fungi association: **The works where endolithic microhabitats were well defined.

| Endolithic habitats in Ca-sulfate-bearing substrates | Nature of Ca-sulfate substrates / Locality | Algae | Fungal hyphae | Proto-lichens* | Cyanobacteria | Heterotrophic bacteria | References** |
|---|---|---|---|---|---|---|---|
| Cryptoendolithic | Gypsum/anhydrite crusts on soil surface / Tarapacá | ■ | ■ | ■ | ■ | ■ | Wierzchos et al. (2011), Vítek et al. (2013) |
| Cryptoendolithic | Gypsum+anhydrite crusts on soil surface / Salar Navidad | | ■ | ■ | ■ | | Culka et al. (2017) |
| Cryptoendolithic | Gypcrete / Cordon de Lila | ■ | | | ■ | | Wierzchos et al. (2015) |
| Chasmoendolithic | Gypsum crust on the surface of rhyolite / Tilocalar | | | | | ■ | DiRuggiero et al. (2013) |
| Hypoendolithic | Gypsum/anhydrite crusts on soil surface / Tarapacá | | | ■ | | | Wierzchos et al. (2011) |
| Hypoendolithic | Gypcrete / Cordon de Lila | | | | ■ | ■ | Wierzchos et al. (2015) |
| Crypto-, Chasmo- and Hypoendolithic | Gypcrete / Cordon de Lila | | | | ■ | ■ | This work |

**Fig. 1.**

[Figure]